# Sustainable Dengue Prevention and Management: Integrating Dengue Vaccination Strategies with Population Perspectives

**DOI:** 10.3390/vaccines12020184

**Published:** 2024-02-12

**Authors:** Asrul Akmal Shafie, Edson Duarte Moreira, Gabriela Vidal, Alberta Di Pasquale, Andrew Green, Rie Tai, Joanne Yoong

**Affiliations:** 1Discipline of Social and Administrative Pharmacy, School of Pharmaceutical Science, Universiti Sains Malaysia, Gelugor 11800, Malaysia; aakmal@usm.my; 2Associação Obras Sociais Irmã Dulce Hospital Santo Antônio and Oswaldo Cruz Foundation, Bahia CEP 40.415-006, Brazil; edson.moreira@fiocruz.br; 3Argentinian Infectious Diseases Society, Buenos Aires C1085, Argentina; gabriela.vidal.sgan@gmail.com; 4Regional Medical Affairs Vaccines, Growth and Emerging Markets, Takeda Pharmaceuticals International AG Singapore Branch, Singapore 018981, Singapore; alberta.di-pasquale@takeda.com (A.D.P.); andrew.green@takeda.com (A.G.); 5Vista Health Pte Ltd., Singapore 059413, Singapore; rtai@vista.health; 6Research For Impact, Singapore 159964, Singapore

**Keywords:** dengue, vaccine, population views, Latin America, Asia-Pacific

## Abstract

The GEMKAP study (2023) unveiled consistent knowledge, attitude, and practice (KAP) levels across Asia-Pacific (APAC) and Latin America (LATAM) countries regarding dengue, with variations in the willingness to vaccinate. Despite an overall KAP parity, the disparities within and between the countries indicated the need for both overarching and tailored strategies. Population-wide gaps in dengue awareness result in suboptimal vaccination priorities and preventive measures. This commentary delves into identifying the drivers and barriers for implementing a multi-pronged dengue prevention and management program, emphasizing the pivotal role of vaccination alongside education and vector control. Drawing on expert interviews in APAC and LATAM, informed by the Consolidated Framework for Implementation Research (CFIR), four key themes emerged: prioritizing and continuously advocating for dengue on national health agendas, fostering stakeholder collaboration, incorporating population perspectives for behavioral change, and designing sustainable dengue prevention and management programs. Successful implementation requires evidence-based decision making and a comprehensive understanding of population dynamics to design adaptive education tailored to diverse population views. This commentary provides actionable strategies for enhancing dengue prevention and management, with a pronounced emphasis on dengue vaccination, advocating for a holistic, population-centric approach for sustained effectiveness.

## 1. Introduction

Dengue is a major public health concern, considered among the most prevalent vector-borne viral diseases [1]. Half of the world’s population is at risk of dengue and 100 to 400 million dengue infections are recorded annually [2]. It is especially prevalent in tropical and subtropical climates, including countries in Asia-Pacific (APAC) and Latin America (LATAM) [3]. APAC alone accounts for about 70% of dengue’s global disease burden. Likewise in LATAM, dengue is hyperendemic and its incidence continues to rise [4]. With higher temperatures, precipitation, and longer periods of droughts due to climate changes, the World Health Organization (WHO) and Pan-American Health Organization (PAHO) have warned of escalating cases of dengue infections worldwide and the need for countries to review their preparedness and response plans [2,5]. Despite the severity of dengue, population-level data on the knowledge, attitudes, and practices (KAP) towards dengue are not readily available or continuously considered. This hinders the design of effective dengue prevention and management programs tailored to each population’s unique needs. Considering population views in the design of a health program enhances its contextual relevance and cultural compatibility [6,7]. Understanding the unique behavior change determinants of the population can encourage participation in personal preventive measures, including vaccination and vector control activities, more effectively. To account for evolving perspectives over time, KAP data should be continuously collected. Through an iterative cycle, programs can be adapted and refined to remain relevant in an ever-changing society and disease landscape [7]. Moreover, collecting population KAP regularly and systematically enables baseline comparisons for monitoring and evaluating the program success. 

Given the growing threat of dengue, effective dengue prevention and management programs that resonate with the population are critical to elicit community efforts in dengue prevention. The published “Knowledge, Attitudes, and Practices toward Dengue Fever, Vector Control, and Vaccine Acceptance Among the General Population in Countries from Latin America and Asia-Pacific: A Cross-Sectional Study (GEMKAP)” [8] identified the key factors influencing populations’ willingness to vaccinate, including the essential need for accessibility and affordability of vaccines. Additionally, the significance of trusted stakeholders, particularly healthcare professionals, was highlighted as crucial for providing further education on the safety and efficacy of vaccines. 

Building on the GEMKAP study, this commentary takes the next step to identify the drivers and barriers of the APAC and LATAM health systems for implementing a national dengue management and prevention program that leverages insights from the general population. To achieve this, insights and perspectives of international and regional experts were collected through interviews, guided by the Consolidated Framework for Implementation Research (CFIR) [9]. Based on the insights collected, the commentary presents recommendations for the effective design and implementation of dengue prevention and management programs.

## 2. Summary of the GEMKAP Study

The GEMKAP study [8] reported that the KAP levels were similar across both APAC and LATAM countries—knowledge (48%, standardized, 0–100% scale) and practice (44%) levels were low, while attitude (66%) levels were moderate. A willingness to vaccinate against dengue was higher in LATAM (59%) than in APAC (40%).

Low-to-moderate KAP levels were attributed by several common factors across the countries. For knowledge levels, 63% were not aware of the different dengue serotypes and 27% were unsure as to the status of registration of a dengue vaccine [8]. Additionally, only 69% of the study population was aware that dengue may present as a severe disease [8]. This led to lower levels in attitudes and practices, with only 18% prioritizing dengue vaccination over other optional vaccines and only an average of 5.8 out of 10 dengue prevention measures being taken. Lastly, only 64% trusted their Ministry of Health to register medically safe vaccines. These factors are critical to address, given that the top driver of a willingness to vaccinate was the desire to be protected against dengue, and the key barriers were concerns about vaccine safety and efficacy. Nevertheless, KAP variations existed between and within the countries. For example, APAC countries showed a higher confidence in their government’s dengue response (APAC: 40%; LATAM: 21%). Religious leaders were especially influential towards health behaviors (including vaccination) in Indonesia and Malaysia, but not in other countries. 

The population consistently preferred a multi-pronged dengue prevention and management program. Therefore, in the context of this commentary, a dengue management program will refer to one that concurrently comprises dengue vaccination, education, and vector control initiatives.

## 3. Methodology

This commentary was developed from interviews with six experts across Argentina (G.V.), Brazil (E.D.M.J.), Indonesia (A.G.), Malaysia (A.A.S.), and Singapore (J.S.Y.Y., A.D.P.) to represent the insights and experiences in APAC and LATAM. The experts included academia and industry representatives who were recruited through purposive sampling. As this commentary builds on the GEMKAP study, four of the experts recruited (E.D.M.J., A.A.S., J.S.Y.Y., A.D.P.) were authors of the GEMKAP study. Additionally, two new experts (G.V., A.G.) were recruited to provide Argentinian and Indonesian insights for more representative APAC and LATAM perspectives. The experts were selected for their expertise across health systems and policy, behavioral economics, and clinical science in vaccines. They had a proficient understanding of dengue disease control, surveillance, and vaccination program implementation. Their varied expertise served to provide a diverse view of the potential drivers and barriers faced when implementing a population-based dengue prevention and management program in APAC and LATAM. 

One hour online interviews were conducted with the experts and were moderated using a discussion guide (Appendix A), which had been developed based on the CFIR, a theory-based framework that systematically assesses drivers and barriers for the successful implementation of an intervention [9]. The CFIR was only used in the development of the discussion guide to ensure systematic and comprehensive data collection (i.e., the insights were collected for all CFIR domains and constructs). The thematic analysis was kept inductive without referencing the CFIR to favor insight generation.

The data from the transcripts was extracted and the data summary was discussed and refined with the experts as a group. Following the data analysis using a content analysis template, four key themes were identified. These themes were supported by the published literature, the GEMKAP study findings, and country or regional case examples to ensure a comprehensive commentary. 

## 4. Discussion

Four key themes surfaced from the analysis of the expert discussions, which identified the drivers and barriers for a successful dengue prevention and management program. These included the need for health systems to (1) continuously prioritize dengue on the national health agenda, (2) ensure collaboration and coordination among all stakeholders, (3) incorporate population views into educational programs to elicit behavioral change, and (4) design these programs for sustainability. These findings served as an additional perspective for dengue prevention and management programs in general and were not specific to a particular strategy or country. However, where mentioned by experts, relevant country-specific examples were used to support each theme.

### 4.1. Continuously Prioritize Dengue on the National Health Agenda

This theme delves into (a) where dengue currently stands in national health agendas, (b) the hidden impact of dengue, (c) the barriers to prioritizing dengue, (d) the strategies to overcome these barriers, and (e) how the agenda can be effectively disseminated to the population.

The GEMKAP study identified governments as having the greatest influence on healthcare decision making among the population [8]. Governments set the national health agenda and prioritize disease areas and prevention methods. Unfortunately, preventing and controlling infectious diseases, including mosquito-borne diseases such as dengue, are not often prioritized by governments, especially when the number of cases subsides during periods without outbreak. From the GEMKAP study, only 23% of respondents believed that their government was well prepared to combat dengue and 28% believed that their government was responding appropriately to the dengue challenge [8]. This illustrates the people’s lack of satisfaction toward governments and their desire for local leaders to prioritize and lead dengue prevention efforts. Additionally, a lack of focus on strengthening health systems can contribute to the re-emergence and spread of pathogens, worsening the disease burden among the population [10]. Furthermore, dengue’s disease burden is expected to increase without a continuously implemented dengue prevention and management strategy, especially given climate change and urbanization [3].

Dengue has a significant impact on individual lives, households, and healthcare systems. With 96 million symptomatic cases recorded annually across the globe, health facilities often see rapid patient influx during dengue outbreaks, causing shortages in infrastructure and staff, especially in low-resource settings [11,12]. Dengue goes beyond its severe clinical manifestations, significantly impacting an individual’s financial, social, and psychological well-being [12]. While the direct medical cost of dengue is considerable, the economic burden of dengue is primarily driven by indirect costs, with the largest proportion arising from factors such as a loss of productivity [13]. Critically, dengue is often misdiagnosed as other febrile illnesses and is underreported even in countries where reporting is mandatory [14,15]. Therefore, dengue’s actual clinical and economic burden remains underestimated and is even greater than what the current available data suggests [16]. The lack of robust dengue surveillance contributes to missed or delayed reporting and response, potentially worsening the disease outcomes [17]. The PAHO reported substantial heterogeneity in the capacity of dengue surveillance systems within the Americas in terms of laboratory testing, data reporting, and case management. Although some countries may have the capacity to control and manage dengue cases successfully, their resources are often stressed in times of dengue outbreaks [17].

Several reasons contribute to the government’s lower priority towards dengue prevention, including dengue’s cyclical outbreaks, relatively lower mortality and morbidity rates compared to other infectious diseases, and challenges in developing and deploying dengue vaccines [14]. Instead, governments prioritize funds towards more urgent diseases with high morbidity and mortality rates, leaving dengue’s impact somewhat ‘hidden’.

To support effective policymaking to integrate vaccinations into dengue prevention and management programs, access to robust and objective evidence, including epidemiological, economic, clinical, and real-world evidence, is required. Such evidence should be generated by various stakeholders, including academia, healthcare professionals, and the pharmaceutical industry. Strong dengue surveillance systems, including reliable and timely diagnoses and data reporting, are essential for robust epidemiological measurements. These epidemiological outputs must showcase the impact of dengue even during periods without outbreak and highlight the potential escalation in the disease burden if dengue prevention weakens. Consistently monitoring dengue epidemiology before and after introducing the dengue vaccine on a large scale can help demonstrate its effectiveness, especially on the dengue burden and prevalence across serotypes [18]. Furthermore, economic evidence showcasing the cost-effectiveness and societal impact of consistently implementing dengue vector control and vaccination programs is crucial for gaining government buy-in [19]. To complete the evidence requirements, emphasizing patient voices with real-life stories can shed light on the individual-level burden of dengue. Concerning the dengue vaccine, larger and longer-term clinical and real-world evidence is expected to reassure decision makers about the safety and efficacy of new vaccines, addressing the limitations identified with previous vaccines. Understanding the complexities of dengue disease and the vaccine can support policymakers in issuing evidence-based implementation guidelines. This underscores the need for a multi-pronged approach to dengue prevention and management, acknowledging the absence of a one-size-fits-all solution. It is critical that dengue vaccination is accompanied by vector control efforts, where neither negates nor detracts from the need for the other [18].

To overcome funding limitations, weaving dengue strategies into broader infectious disease and preventive health agendas is recommended. Leveraging existing health programs’ infrastructure and networks can increase economies of scale. For example, vector control measures for dengue will also help in the prevention of other arboviruses, such as Chikungunya and Zika [18]. Cost-effective dengue strategies or alternative financing mechanisms can support vaccine uptake, allowing governments to prioritize and fund long-term programs. Many countries have existing dengue education and vector control programs that can be revived and strengthened alongside a vaccination program. 

Finally, to mobilize the population towards dengue prevention, governments should guide the population to practice personal preventive measures, including vaccination and vector control. This is critical, given that the GEMKAP study found the practice levels to be the lowest (44%), as compared to knowledge (66%) and attitudes (48%) [8]. Encouraging the practice of preventive measures can be achieved by consistently disseminating a narrative on the importance of combating dengue when setting a national health agenda. Such messaging should be grounded in robust and objective evidence on the intricate nature of dengue disease and the associated vaccine. Leveraging trusted intermediary stakeholders to drive this narrative can help create behavior change at the individual level. Successfully doing so will require collaboration across all stakeholders, such that a single congruous narrative is disseminated nationwide.

### 4.2. Ensure Collaboration and Coordination among All Stakeholders

With the integration of dengue vaccination into existing education and vector control initiatives, the multi-faceted nature of a multi-pronged dengue prevention and management program implemented at the national level requires strong collaboration between and within stakeholders. The successful implementation of such a program is dependent on having well-defined roles and responsibilities for each stakeholder and developing clear and transparent communication channels within and among stakeholder groups. Figure 1 showcases the key stakeholders of a dengue prevention and management program and illustrates their interconnected nature and reliance in terms of the design and implementation of the program.

As identified from the GEMKAP study [8], each stakeholder can bring unique value and expertise to the design and implementation of a dengue prevention and management program. Government ministries serve as the primary advocates for health initiatives, formulating and disseminating national guidelines to facilitate program implementation by local actors. Managing dengue vaccine approval, procurement, and reimbursement is also critical for ensuring timely and adequate access to new dengue vaccines. The government, in collaboration with intermediary bodies, can also systematically engage with the public, eliciting feedback to adapt and refine dengue prevention and management strategies for optimal efficacy. Technical advisors, comprising clinical, entomology, implementation, and environmental experts, can offer critical insights into the formulation of comprehensive dengue prevention and management approaches. Healthcare professionals, who were identified in the GEMKAP study to be the most trusted stakeholders for health-related information, assume a pivotal role in disseminating information about dengue disease and vaccination [8]. This involves transparent communication about both the advantages and risks associated with the dengue vaccine. Community leaders can mobilize local populations to adopt healthier practices, adhere to dengue prevention initiatives, and encourage vaccine uptake [20]. Beyond continuing innovation in dengue prevention, the industry also plays a role in maintaining access to and facilitating education efforts on vaccines. For example, vaccine manufacturers can provide ongoing medical education programs and ensure the supply of dengue vaccines. As public involvement is integral, the general population should actively shape program preferences and articulate community needs and priorities.

The effectiveness of a dengue prevention and management program heavily relies on how stakeholders collaborate [20]. The collaborative approach should inherently be reciprocal, involving comprehensive feedback from all stakeholders. This is especially crucial when adapting strategies to diverse geographic contexts and specific sub-groups. Some barriers to achieving strong, multi-stakeholder collaboration include a potential lack of alignment with the national priority for dengue prevention and a lack of clear leadership. If these barriers are not overcome, differences in stakeholder priorities can ultimately jeopardize the program’s success. As a multi-pronged dengue prevention and management program comprises various initiatives implemented simultaneously by different stakeholders, a lack of coordination can lead to misalignment in the different prongs and cause delays in program implementation. For example, it is critical to drive the message that the introduction of new dengue vaccines does not diminish the need for continued vector control activities [18]. When building a new narrative for new dengue vaccines, it is also important to emphasize its differences in comparison to previous dengue vaccines. While leveraging different stakeholders to communicate to the population can help to tailor messages to local contexts, the key narratives should remain consistent [18]. This is important to prevent misinformation, which may generate hesitancy in vaccinating against dengue and participation in vector control activities [21]. 

Fostering effective communication among all stakeholders is pivotal for successful dengue prevention and management [19]. This has been exemplified in Malaysia [22,23] and Singapore [24], where a unified group of multi-sectorial stakeholders formulated clear strategies and oversaw dengue prevention and management programs. This ensured that all stakeholders had aligned objectives and delivered consistent messaging. Since 2011, the Malaysian Ministry of Health, in collaboration with six other ministries, has put into action the “Integrated Management Strategy” for the implementation of dengue prevention and control activities. These efforts have resulted in a remarkable reduction in dengue incidence nationwide within just three months [22]. Similarly in Singapore, their dengue task force comprises more than 20 stakeholders across ministries, research and academia, community structures, councils, and private institutions [24]. The formation of the dengue task force enhanced the coordination of dengue control efforts between task force members who meet regularly and monthly in times of dengue outbreak [24].

### 4.3. Incorporate Population Views to Elicit Behavioral Change

To design an effective health program that resonates with the public, health program designers and implementers must have a good understanding of population views. The success of vector control and vaccination programs is largely dependent on an individual’s agency to act. Therefore, in designing a dengue prevention and management program, understanding population views and customizing interventions accordingly would influence action more effectively [20]. Furthermore, understanding and applying the key factors that elicit behavior change is crucial to boost an adoption of and adherence to dengue prevention activities [20]. The key factors that elicit behavior change include the understanding of the disease and vaccine, perception of the prevention methods, and belief in the effectiveness of practicing the prevention methods. These factors can be categorized using the KAP framework and should be identified and differentiated by countries and sub-groups. The KAP findings can also be leveraged to customize a multi-pronged approach, which includes education, vector control and vaccinations. The key KAP factors identified from the GEMKAP study [8] are listed below, along with their resulting implications when designing a multi-pronged dengue prevention and management program.

Knowledge: There is a lack of awareness of the risk of dengue re-infection, the existence of a dengue vaccine, and a lack of understanding of the safety and efficacy of a vaccine. Education programs can specifically fill these pre-identified knowledge gaps to empower populations to proactively safeguard their health. Additionally, vaccination campaigns should first aim to raise awareness about the severity of dengue and introduce vaccines as one of the solutions to minimize the risk of infections for improved uptake and adherence.Attitude: The overall sentiment regarding the importance of vaccines is positive. However, some individuals express skepticism towards trying a new vaccine due to concerns about potential side effects. Perceptions on the effectiveness of vector control were positive, however confidence in the government’s ability to combat dengue was low. For vaccine education programs, enhancing the understanding of the characteristics and potential side effects of the new dengue vaccine can contribute to a more positive attitude towards it. Governments can initiate and promote vector control and vaccination programs to instill confidence in their ability to combat dengue, which in turn will further increase the practice of prevention.Practice: Common vector control measures, such as discarding standing water and wearing long sleeves, are generally practiced. However, to maximize their full effect, strict adherence is necessary. Regarding a willingness to be vaccinated, inducing behavior change can involve offering incentives for compliance with country regulations and strategically locating vaccine centers in convenient and accessible locations.

In addition to these key factors, communication channels and sources for dengue prevention and vaccine messaging must be tailored to meet the population’s needs. Different countries and sub-populations may prefer specific communication approaches. While the GEMKAP study identified search engines as the preferred communication channel consistently across countries, trusted stakeholders for health messaging differed [8]. For example, the Singaporean population trusted communication via the government up to three times more than LATAM populations [8]. 

Applying population views to customize the design of a multi-pronged program will increase the adoption of and adherence to dengue prevention activities. Additionally, integrating population perspectives, alongside epidemiological contexts, into the design of flexible health programs is not only effective but also aligns with the goal of ensuring equitable access to dengue prevention measures, including vaccines. Stratifying perspectives to pinpoint the needs of minority sub-populations ensures that their unique requirements are not overshadowed by the general needs of the wider population [20].

### 4.4. Design for a Sustainable Health Program

Policymakers, funders, and community partners are increasingly interested in the sustainability of effective health intervention programs [25]. In the context of dengue prevention, it is recommended to establish a dengue task force or alliance alongside national sustainability efforts to ensure the long-term viability of such programs [15]. This alliance should bring together diverse stakeholders across various sectors to share expertise and resources to ultimately achieve a meaningful and significant reduction in symptomatic dengue incidence. This prioritization of dengue prevention and management should be sustained at both national and international levels through the following strategies.

Continuously integrating evolving population perspectives and adapting health programs to align with the prevailing epidemiology and population needs [6,7,20].Strengthening the capacity of community health workers and healthcare professionals to implement dengue prevention programs, enhance disease surveillance capabilities, implement vector control initiatives, encourage vaccine uptake, and elevate overall disease management [20,26].Facilitating the exchange of good practices, research findings, and technological advancements in dengue prevention and control through knowledge and resource sharing [15,27,28].Advocating for increased financial support from governments, international organizations, and private sector partners to bolster dengue prevention and control efforts, including equitable access to dengue vaccines [20,28].Cultivating collaboration between research institutions, academia, and industries to advance scientific understanding and innovation in dengue control [15,28].Promoting public awareness and education regarding dengue prevention measures, symptoms, and the benefits of vaccination and vector control [15,20].Establishing robust epidemiological monitoring and economic evaluation mechanisms to track the impact of interventions, disseminating findings to relevant decision makers and stakeholders to enable evidence-driven decision making [17,28].

In an increasingly interconnected world, infectious diseases like dengue transcend national borders. Climate change and urbanization further exacerbate the spread of dengue, potentially exposing previously unaffected populations [18,20]. To sustain this elevated prioritization, the dengue alliance must engage in comprehensive educational initiatives targeting the current and next generation. This includes school programs and community engagement efforts to cultivate future leaders in public health and instill a culture focused on dengue prevention.

There are some limitations to this study. Firstly, the sample of six experts interviewed may not be fully representative of the diverse APAC and LATAM health systems. Therefore, future research should consider a more comprehensive sample of experts across various geographies and expertise. Secondly, the experts interviewed were also authors of this commentary, potentially introducing bias in the narrative. To minimize bias, the commentary was supported by the published literature, where applicable. Thirdly, the use of the CFIR was limited to only the discussion guide development. Future studies on dengue prevention program implementation should consider fully leveraging the CFIR to identify implementation drivers and barriers specific to their contexts of interest.

## 5. Conclusions

To effectively combat dengue, collaboration and coordination among all national and international stakeholders is essential. The following are a set of next steps.

Collect robust and objective evidence from various stakeholders to support evidence-based decision making when prioritizing dengue prevention on the national health agenda.Establish national and international dengue task forces with stakeholders from different sectors to collaborate and share good practices, capabilities, and resources.Continuously collect and incorporate population views to facilitate the design and adaptation of dengue prevention programs that fit the needs of the people.Ensure long-term sustainability of dengue programs by demonstrating their impact through surveillance systems that monitor and evaluate their success metrics.

In a world where the containment of infectious diseases is no longer just a national-level problem, national and international communities must come together to create a space for open dialogue and collaboration where population views are fundamental to achieve a significant reduction in symptomatic dengue incidence. 

## Figures and Tables

**Figure 1 vaccines-12-00184-f001:**
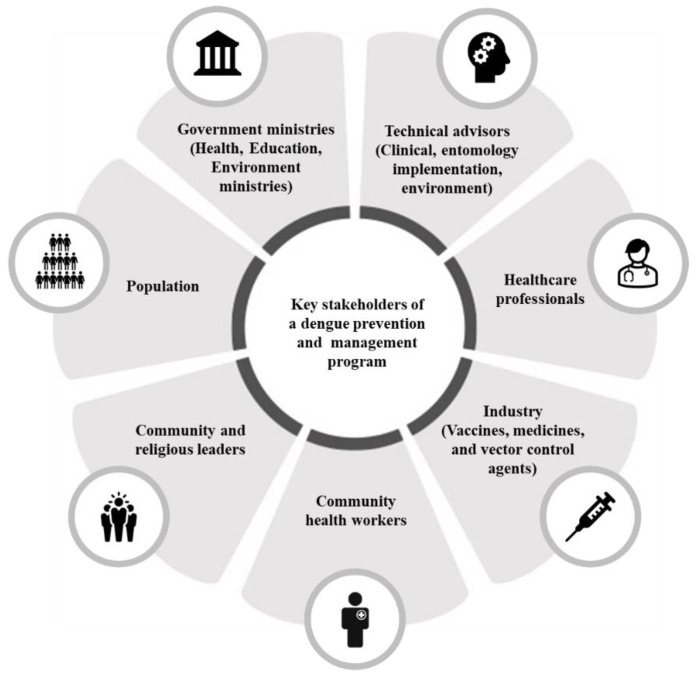
Key stakeholders in the design and implementation of a dengue prevention and management program.

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
