# Peer review of "Sustainable Dengue Prevention and Management: Integrating Dengue Vaccination Strategies with Population Perspectives"

_vaccines, 2024, doi:10.3390/vaccines12020184_

Round 1

Reviewer 1 Report

Comments and Suggestions for Authors

The success of a vaccine effort will depend on vaccine efficacy and safety and the communication of the results to stakeholders and funders.  That means that the results must be quantifiable and reliable. That, in turn, depends on establishing accurate numbers of cases before vaccination efforts and the number after. To accomplish that, a well planned surveillance effort with reliable diagnosis with timely laboratory support and communication and interpretation of the results is required to know how many true dengue cases there are in a given area.  The authors only mention this briefly on lines 320-321 saying, "Establishing robust monitoring and evaluation mechanisms to track the impact of interventions, enabling evidence-driven decision-making." Establishing a confirmed dengue diagnosis means that a differential diagnosis is needed to avoid false negatives by attributing a febrile clinical presentation as malaria, as often the case in Africa, or false positives by attributing them to dengue when, in fact, these febrile cases are due to other arboviruses such as Oropouche virus infection in tropical South America. Gaps in surveillance can lead to underreporting in situations such as those in India where ill patients may go to private hospitals and the dengue diagnosis may not be confirmed nor case numbers reported to national health authorities. In addition to the valid points that the authors clearly make, the success of vaccination efforts turns on surveillance, laboratory support to confirm the dengue diagnosis, timely reporting of the results to epidemiologists and then on to decision-makers and other stake-holders. The ability to accomplish this is highly variable among countries, or even within some countries, around the dengue-endemic world. 

Are there any lessons to be learned from the successful smallpox eradication program or the struggling polio eradication program in the two remaining endemic countries, Afganistan and Pakistan, that are relevant to dengue elimination through vaccination?

There are several relatively new, pilot approaches to dengue virus vector (mainly Aedes aegypti) control, some of which have shown success in reduction of dengue cases in the field. Will widespread adoption of these approaches, with corresponding reduction of reported dengue cases, support or detract from vaccination efforts? 

Author Response

Dear Reviewer, 

Thank you for reviewing and providing thoughtful comments on the manuscript. We have provided a reply to your comments and updated the manuscript accordingly in track-changes. The comment number can be referenced through the speech bubble comments made in the updated manuscript. 

Kind regards, 

Rie, on behalf of all of the authors

Reviewer 2 Report

Comments and Suggestions for Authors

The current manuscript is a commentary that discusses needs and challenges in implementing effective dengue prevention and management programs based around vaccination, vector control and education.

Authors used a CFIR framework to identify barriers/facilitators to effective dengue program implementation and identified 4 themes that governments should focus on: advocacy, stakeholder collaboration, incorporating population perspectives and sustainability.  The CFIR aims to predict or explain barriers and facilitators to implementation effectiveness and provides a list of constructs to consider. This was a very interesting and potentially robust approach to identify practical areas to support successful dengue program implementation.

One strength of the commentary was the tie in with the GEMKAP study and strong case made for why KAP data is needed for effective dengue programs. The authors link the GEMKAP study finding with each of the 4 themes identified in their CFIR-framework analysis.

Major Comments

The commentary does not clearly state purpose and study question. For example, purpose is described as: identifying barriers/facilitators to integrating KAP data in dengue prevention and management using the CFIR framework (thus tying in with the GEMKAP study), later on as barriers/drivers to implementing a 3-pronged dengue program, and in the discussion this broadens to general characteristics of successful dengue programs (L122).   A critical review of language and thematic consistency will help make the authors' points more clear to the reader. I also feel that narrowing the focus will allow the authors to go into more depth rather than breadth in suggestions for effective dengue prevention and control.

Methods say that CFIR framework analysis was used to identify barriers/facilitators, based on expert interview discussions.  Method of recruiting experts needs more description and supplementary information on expert interview discussions was not included in material for review. For example, further information on who the experts consulted were and how they were selected, and if any experts outside the authors themselves were engaged.  If authors are also the experts, there may be biases in data collection, analysis and conclusions which should be dealt with in the discussion.

Discussion could benefit from explaining how findings link with CFIR domain and constructs, since this was the methods and analysis used.

Commentary is poorly referenced through much of the discussion. This may be because the text describes expert opinion, but many of the points raised for example leveraging existing program infrastructure, integrated control strategies and engaging and mobilizing communities are concepts that are well discussed in other documents, including the WHO Global Vector Control Response.  Thus, it would be useful if the key points identified from the expert interviews and CFIR framework analysis were put into context of broader published literature.

(L236-242) I found the discussion of how to balance the narratives around vaccination, education and vector control very interesting. I would have liked to this discussed more in the context of the GEMKAP study findings and how KAP can inform engagement around all 3 prongs.

Section 4.3 discusses key factors identified in the GEMKAP study and provide further explanation and context based on the current analysis. This section provides a clear rationale for the need to integrate KAP data into control programs, and why KAP data, collected regularly and systematically can help design relevant programs, encourage participation in vector control and vaccination.

Minor comment

L4 - 4 of the 6 authors on the commentary were also authors of the GEMKAP study including first, last and corresponding. Some clear statement of this connection should be made in the introduction or methods.

L53 - missing comma after activities.

L81 - needs additional descriptor of KAP gaps (i.e. population KAP gaps in which categories?) - first paragraph says KAP levels were similar across countries, second paragraph says gaps were apparent across countries.

L137 - Maybe rephrase, or link with citation - many countries, including Singapore, have strong and proactive dengue/arboviral disease control programs.

L146 - The cited study does not support the statement that health systems are "overwhelmed", rather study authors conclude that "resulted in delayed diagnoses for dengue and other diseases but was not reported to have an impact on care and treatment provision."

L159 - what kind of evidence is needed from academia, industry, healthcare?

L250 - Suggest toning down this statement - while the efforts of Malaysia are certainly substantial and should be applauded, the 60% reduction is stated but not well explained in the reference cited (60% compared with what? - this could have been a seasonal or surveillance effect)

Author Response

(The authors gave the same response as above.)
